# Gut bacterial aggregates as living gels

**Brandon H Schlomann[1,2], Raghuveer Parthasarathy[1]\***

[1]Department of Physics, Institute of Molecular Biology, and Materials Science Institute, University of Oregon, Eugene, United States; [2]Department of Physics and Department of Molecular and Cell Biology, University of California, Berkeley, Berkeley, United States

**Abstract** The spatial organization of gut microbiota influences both microbial abundances and host-microbe interactions, but the underlying rules relating bacterial dynamics to large-scale structure remain unclear. To this end, we studied experimentally and theoretically the formation of three-dimensional bacterial clusters, a key parameter controlling susceptibility to intestinal transport and access to the epithelium. Inspired by models of structure formation in soft materials, we sought to understand how the distribution of gut bacterial cluster sizes emerges from bacterial-scale kinetics. Analyzing imaging-derived data on cluster sizes for eight different bacterial strains in the larval zebrafish gut, we find a common family of size distributions that decay approximately as power laws with exponents close to $-2$, becoming shallower for large clusters in a strain-dependent manner. We show that this type of distribution arises naturally from a Yule-Simons-type process in which bacteria grow within clusters and can escape from them, coupled to an aggregation process that tends to condense the system toward a single massive cluster, reminiscent of gel formation. Together, these results point to the existence of general, biophysical principles governing the spatial organization of the gut microbiome that may be useful for inferring fast-timescale dynamics that are experimentally inaccessible.

**\*For correspondence:**
raghu@uoregon.edu

**Competing interests:** The authors declare that no competing interests exist.

## Introduction

The bacteria inhabiting the gastrointestinal tracts of humans and other animals make up some of the densest and most diverse microbial ecosystems on Earth (*Lloyd-Price et al., 2017*; *Sender et al., 2016*). In both macroecological contexts and non-gut microbial ecosystems, spatial organization is well known to impact both intra- and inter-species interactions (*McNally et al., 2017*; *Tilman and Kareiva, 2018*; *Weiner et al., 2019*). This general principle is likely to apply in the intestine as well, and the spatial structure of the gut microbiome is increasingly proposed as an important factor influencing both microbial population dynamics and health-relevant host processes (*Tropini et al., 2017*; *Donaldson et al., 2016*). Moreover, recent work has uncovered strong and specific consequences of spatial organization in the gut, such as proximity of bacteria to the epithelial boundary determining the strength of host-microbe interactions (*Vaishnava et al., 2011*; *Wiles et al., 2020*), and antibiotic-induced changes in aggregation causing large declines in gut bacterial abundance (*Schlomann et al., 2019*). Despite its importance, the physical organization of bacteria within the intestine remains poorly understood, in terms of both in vivo data that characterize spatial structure and quantitative models that explain the mechanisms by which structure arises.

Recent advances in the ability to image gut microbial communities in model animals have begun to reveal features of bacterial spatial organization common to multiple host species. Bacteria in the gut exist predominantly in the form of three-dimensional, multicellular aggregates, likely encased in mucus, whose sizes can span several orders of magnitude. Such aggregates have been observed in mice (*Moor et al., 2017*), fruit flies (*Koyama et al., 2020*), and zebrafish (*Jemielita et al., 2014*; *Schlomann et al., 2018*; *Wiles et al., 2016*; *Schlomann et al., 2019*; *Wiles et al., 2020*), as well as

**eLife digest** The human gut is home to vast numbers of bacteria that grow, compete and cooperate in a dynamic, densely packed space. The spatial arrangement of organisms – for example, if they are clumped together or broadly dispersed – plays a major role in all ecosystems; but how bacteria are organized in the human gut remains mysterious and difficult to investigate.

Zebrafish larvae provide a powerful tool for studying microbes in the gut, as they are optically transparent and anatomically similar to other vertebrates, including humans. Furthermore, zebrafish can be easily manipulated so that one species of bacteria can be studied at a time.

To investigate whether individual bacterial species are arranged in similar ways, Scholmann and Parthasarathy exposed zebrafish with no gut bacteria to one of eight different strains. Each species was then monitored using three-dimensional microscopy to see how the population shaped itself into clusters (or colonies).

Schlomann and Parthasarathy used this data to build a mathematical model that can predict the size of the clusters formed by different gut bacteria. This revealed that the spatial arrangement of each species depended on the same biological processes: bacterial growth, aggregation and fragmentation of clusters, and expulsion from the gut.

These new details about how bacteria are organized in zebrafish may help scientists learn more about gut health in humans. Although it is not possible to peer into the human gut and watch how bacteria behave, scientists could use the same analysis method to study the size of bacterial colonies in fecal samples. This may provide further clues about how microbes are spatially arranged in the human gut and the biological processes underlying this formation.

in human fecal samples (*van der Waaij et al., 1996*). However, an understanding of the processes that generate these structures is lacking.

The statistical distribution of object sizes can provide powerful insights into underlying generative mechanisms, a perspective that has long been applied to datasets as diverse as galaxy cluster sizes (*Hansen et al., 2005*), droplet sizes in emulsions (*Lifshitz and Slyozov, 1961*), allele frequency distributions in population genetics (*Neher and Hallatschek, 2013*), immune receptor repertoires (*Nourmohammad et al., 2019*), species abundance distributions in ecology (*Hubbell, 1997*), protein aggregates within cells (*Greenfield et al., 2009*), and linear chains of bacteria generated by antibody binding (*Bansept et al., 2019*). A classic example of the understanding provided by examining size distributions comes from the study of gels. In polymer solutions, random thermal motion opposes the adhesion of molecules, resulting in cluster size distributions dominated by monomers and small clusters. Gels form as adhesion strength increases, and monomers stick to one another strongly enough to overcome thermal motion and form a giant connected cluster that spans the size of the system. This large-scale connectivity gives gels their familiar stiffness as seen, for example, in the wobbling of a set custard. Theoretical tools from statistical mechanics and the study of phase transitions relate the cluster size distribution to the inter-monomer attraction strength and the temperature (*Krapivsky et al., 2010*). In addition to providing an example of the utility of analyzing size distributions, gels in particular are a ubiquitous state of matter in living systems whose physical properties influence a wide range of activities such as protection at intestinal mucosal barriers (*Datta et al., 2016*) and transport of molecules through amyloid plaques (*Woodard et al., 2014*).

Motivated by these analogies, we sought to understand the distribution of three-dimensional bacterial cluster sizes in the living vertebrate gut, aiming especially to construct a quantitative theory that connects bacterial-scale dynamics to global size distributions. Such a model could be used to infer dynamical information in systems that are not amenable to direct observation, such as the human gut. Identifying key processes that are conserved across animal hosts would further our ability to translate findings in model organisms to human health-related problems. At a finer level, validated mathematical models could be used to infer model parameters of specific bacterial species of interest, for example pathogenic invaders or deliberately introduced probiotic species, by measuring their cluster size distribution.

We analyzed bacterial cluster sizes obtained from recent imaging-based studies of the larval zebrafish intestine (*Schlomann et al., 2018*; *Schlomann et al., 2019*; *Wiles et al., 2020*). As detailed

below, we find a common family of cluster size distributions with bacterial species-specific features. We show that these distributions arise naturally in a minimal model of bacterial dynamics that is supported by direct observation. The core mechanism of this model involves growth together with a fragmentation process in which single cells leave larger aggregates. Strikingly, this process can be mapped exactly onto population genetics models of mutation, with cluster size analogous to allele frequency and single-cell fragmentation analogous to mutation. The combination of growth and fragmentation generates size distributions with power law tails, consistent with the data. This process also maps onto classic network models of preferential attachment (*Barabasi and Albert, 1999*). Further, we show that cluster aggregation can generate an overabundance of large clusters through a process analogous to the sol-gel transition in polymer and colloidal systems, leading to plateaus in the size distribution that are observed in the data. These features of the size distribution are robust to the inclusion of a finite carrying capacity that limits growth and cluster loss due to expulsion from the intestine. In summary, we find that gut microbiota can be described mathematically as 'living gels', combining the statistical features of evolutionary dynamics with those of soft materials. Based on the generality of our model and our observations across several different bacterial species, we predict that this family of size distributions is universal across animal hosts, and we provide suggestions for testing this prediction in various systems.

## Results

### Different bacterial species share a common family of broad cluster size distributions in the larval zebrafish intestine

We combined and analyzed previously generated datasets of gut bacterial cluster sizes in larval zebrafish (*Schlomann et al., 2018*; *Wiles et al., 2020*). In these experiments, zebrafish were reared devoid of any microbes, that is 'germ-free', and then mono-associated with a single, fluorescently labeled bacterial strain (*Figure 1A*). After a 24 hr colonization period the complete intestines of live hosts were imaged with light sheet fluorescence microscopy (*Keller et al., 2008*; *Parthasarathy, 2018*; *Figure 1B*). Bacteria were identified in the images (*Figure 1C*) using a previously described image analysis pipeline (*Jemielita et al., 2014*; *Schlomann et al., 2018*). Single bacterial cells and multicellular aggregates were identified separately, and then the number of cells per multicellular aggregate was estimated by dividing the total fluorescence intensity of the aggregate by the mean intensity of single cells (Materials and methods).

In total, we characterized eight different bacterial strains, summarized in *Table 1*. Six of the strains were isolated from healthy zebrafish (*Stephens et al., 2016*) and then engineered to express

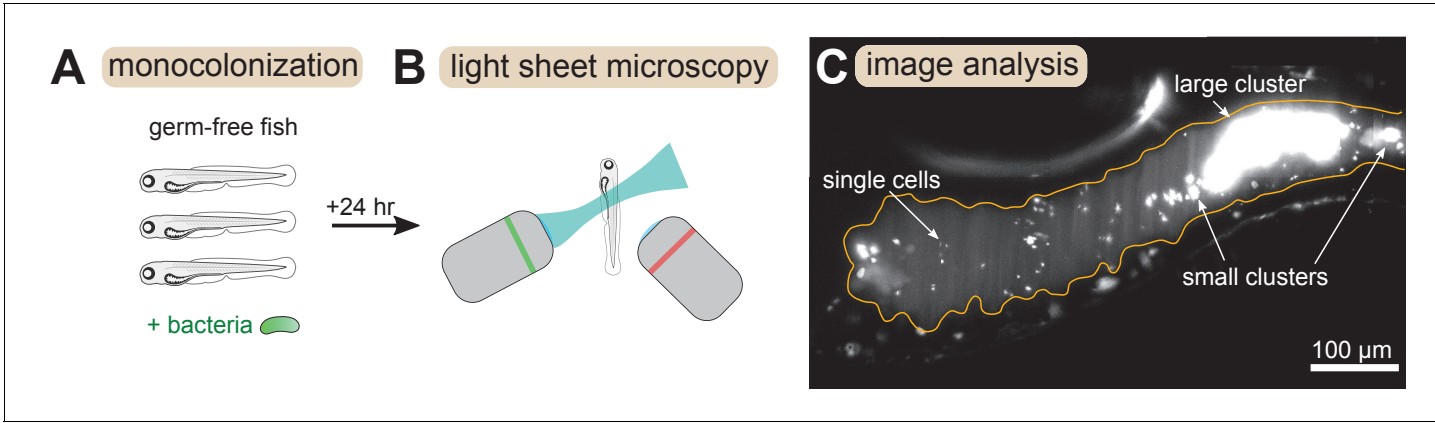

**Figure 1.** Overview of experimental methods. Larval zebrafish were derived germ-free and then monoassociated with single bacterial species (left). After 24 hr of colonization, images spanning the entire gut were acquired with light sheet fluorescence microscopy (middle). An example image of the anterior intestine is shown on the right, with instances of single cells and multicellular aggregates marked. The image is a maximum intensity projection of a 3D image stack. The approximate boundary of the gut is outlined in orange. Sizes of bacterial clusters were estimated with image analysis by separately identifying single cells and multicellular aggregates, and then normalizing the fluorescence intensity of aggregates by the mean single cell fluorescence.

**Table 1.** Summary of cluster data by bacterial strain.

Each row corresponds to one of the bacterial strains included in this study. Entries include strain name, total number of fish colonized with that strain, total number of clusters identified across all fish, and the original publication that the data was pulled from.

| Bacterial strain | Number of fish | Number of clusters | Source publication |
| --- | --- | --- | --- |
| *Aeromonas* ZOR0001 | 6 | 445 | *Schlomann et al., 2018* |
| *Aeromonas* ZOR0002 | 6 | 1901 | *Schlomann et al., 2018* |
| *Enterobacter* ZOR0014 | 18 | 3597 | *Schlomann et al., 2018*; *Schlomann et al., 2019* |
| *Plesiomonas* ZOR0011 | 3 | 223 | *Schlomann et al., 2018* |
| *Pseudomonas* ZWU0006 | 6 | 133 | *Schlomann et al., 2018* |
| *Vibrio* ZOR0036 | 6 | 2430 | *Schlomann et al., 2018* |
| *Vibrio* ZWU0020 Δmot | 11 | 5888 | *Wiles et al., 2020* |
| *Vibrio* ZWU0020 Δche | 11 | 3551 | *Wiles et al., 2020* |

fluorescent proteins (*Wiles et al., 2018*), and two are genetically engineered knockout mutants of *Vibrio* ZWU0020, defective in motility (specifically, knockout of the two-gene operon encoding the polar flagellar motor, *pomAB*, referred to as 'Δmot') and chemotaxis (specifically, knockout of the histidine kinase *cheA2*, referred to as 'Δche'), as described in reference (*Wiles et al., 2020*). The parent strain of these mutants, *Vibrio* ZWU0020, scarcely forms aggregates at all, existing primarily as single, highly motile cells (*Wiles et al., 2016*; *Schlomann et al., 2019*; *Wiles et al., 2020*), and so was excluded from this analysis. All strains are of the phylum Proteobacteria (*Wiles et al., 2018*). A table of all cluster sizes by sample is included in *Figure 2—source data 1*.

We calculated for each bacterial strain the reverse cumulative distribution of cluster sizes, $P(\text{size}>n)$ , denoting the probability that an intestinal aggregate will contain more than $n$ bacterial cells. We computed $P(\text{size}>n)$ separately for each animal (*Figure 2*, small circles) and also pooled the sizes from different animals colonized by the same bacterial strain (*Figure 2*, large circles). There is substantial variation across fish, but the pooled distributions exhibit a well-defined average of the individual distributions. We also computed binned probability densities (*Figure 2—figure supplement 1*), which show similar patterns, but focus our discussion on the cumulative distribution to circumvent technical issues related to bin sizes.

We find broad distributions of $P(\text{size}>n)$ across all strains (*Figure 2*, bottom right panel). For comparison, for each strain we overlay a dashed line representing the power law distribution $P(\text{size}>n) \sim n^{-1}$. This $P(\text{size}>n)$ is equivalent to a probability density of $p(n) \sim n^{-2}$ since the latter is proportional to the derivative of the former. Each strain's cumulative distribution follows a similar power-law-like decay at low $n$, with an apparent exponent in the vicinity of -1, and then becomes shallower in a strain-dependent manner. For example, *Aeromonas* ZOR0002 has a quite straight distribution on a log-log plot (*Figure 2*, top row, middle column), while the distribution of *Enterobacter* ZOR0014 exhibits a plateau-like feature at large sizes (*Figure 2*, top row, right column). The mutant strains *Vibrio* ZWU0020 Δche and Δmot follow qualitatively similar distributions to the native strains (*Figure 2*, bottom row, left and middle columns).

We performed a sensitivity analysis and found that these two key features of the measured distributions—an initial power law-like decay with cumulative distribution exponent close to -1 and a strain-dependent plateau at large sizes—are robust to measurement error in enumeration of cluster sizes. For the initial decay of the distribution, the largest source of error is the misidentification of auto-fluorescent background as single cells. To assess the impact of our single-cell count uncertainty on the distribution, we fit a power law model to clusters sizes up to 100 cells two times: once including single cells and once considering only cells of size in the range 2–100 (*Supplementary file 1*, Materials and methods). In both fits we find cumulative distribution exponents consistent with −1 for most strains. The average exponent tended to decrease mildly when single cells were excluded from the fit (the distribution decayed more slowly), consistent with an over-estimation of the number of single cells, but the shifts were all within uncertainties. Estimates of distribution exponents from

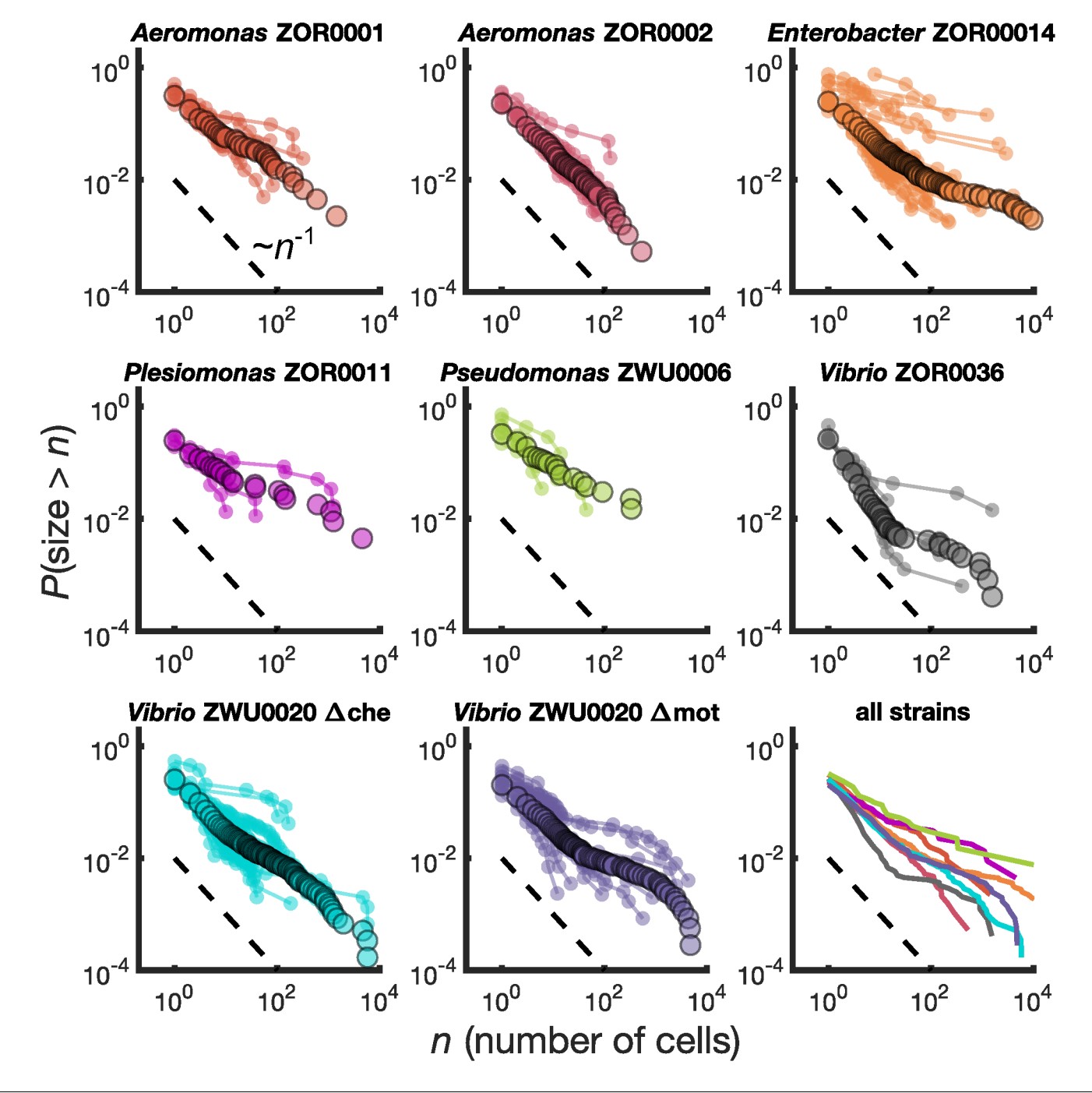

**Figure 2.** Different bacterial species exhibit similar cluster size distributions. Reverse cumulative distributions, the probability that the cluster size is greater than $n$ as a function of $n$, for eight bacterial strains in larval zebrafish intestines. Small circles connected by lines represent the distributions constructed from individual fish. Large circles are from pooled data from all fish. The dashed line represents $P(\text{size} > n) \sim n^{-1}$ and is a guide to the eye. Bottom right panel shows the pooled distributions for each strain as solid lines.

The online version of this article includes the following source data and figure supplement(s) for figure 2:

**Source data 1.** Spreadsheet with all cluster sizes by strain.

**Figure supplement 1.** Cluster size distributions as probability densities.

**Figure supplement 2.** Images of individual $z$-slices showing mild heterogeneity of fluorescence intensity within aggregates.

small sizes can easily be biased (*Clauset et al., 2009*), so we performed our sensitivity analysis with two different methods: a linear fit to $\log P(\text{size} > n)$ vs. $\log n$, and maximum likelihood estimation (Materials and methods). The maximum likelihood estimate gave higher values than line-fitting, but the shifts upon removing single cells were within uncertainties for both methods.

For the large-size plateau, the existence of dim cells in the center of the aggregate, perhaps due to a state of low metabolic activity, would lead to an underestimate of total cluster size. Underestimating the size of large clusters would then result in a less extreme plateau; the plateaus we observe are therefore a lower bound. In cross-sections of large aggregates, we observe mostly homogeneous fluorescence, suggesting that this effect is mild, although small dark regions do occur (*Figure 2—figure supplement 2*). Whether these dark regions correspond to dead or inactive bacteria, mucus, or empty space, is not clear, although we note that small clumps of dead bacteria have been observed in expelled clusters via live/dead staining (*Schlomann et al., 2019*). Regardless of their origin, we conclude that these mild heterogeneities are unlikely to significantly alter the behavior of the size distributions, which span 4 orders of magnitude.

In summary, we find that different bacterial strains, which exhibit a variety of swimming and sticking behaviors (*Wiles et al., 2018*; *Schlomann et al., 2018*), abundances (*Schlomann et al., 2018*; *Wiles et al., 2020*), and population dynamics (*Wiles et al., 2016*; *Schlomann et al., 2019*; *Wiles et al., 2020*), share a common family of cluster size distributions. This observation suggests that generic processes, rather than strain-specific ones, determine gut bacterial cluster sizes. Notably, these distributions are extremely broad, inconsistent with the exponential-tailed distributions found for linear chains of bacteria (*Bansept et al., 2019*). We next sought to understand the kinetics that give rise to our measured cluster size distributions.

## A growth-fragmentation process generates power-law distributions

Previous time-lapse imaging of bacteria in the zebrafish intestine revealed four core processes that can alter bacterial cluster sizes: (1) clusters can increase in size due to cell division, a process we refer to as 'growth'; (2) clusters can decrease in size as single bacteria escape from them, a process we refer to as 'fragmentation' and believe to be linked to cell division at the surface; (3) clusters can increase in size by joining with another cluster during intestinal mixing, a process we refer to as 'aggregation'; and (4) clusters can be removed from the system by transiting along and out of the intestine, a process we refer to as 'expulsion'. The breakup of large clusters into medium ones appears to be rare in our system, so we ignore this process. The single cell fragmentation process we describe conserves cell number and is analogous to the 'chipping' kernel that has been used to describe the breaking off of monomers from the ends of linear polymers (*Krapivsky and Redner, 1996*).

To understand how each of these process affect the distribution of cluster sizes, we used mathematical modeling. We attempted to construct a simple model that encoded these processes and retained salient biological and physical features. In our model, the relevant variable is a list of all cluster sizes, or equivalently, a list of the number of clusters of each size. Clusters can change size according to four reactions that correspond to each of the four processes listed above. There is no explicit spatial dependence in this model, but aspects of spatial structure, such as the fact that some cells in a cluster are confined to the center while others are on the surface, can be modeled by choosing how the rates of reactions depend on cluster size, as discussed below. We assume, however, that growth rates are the same for all cells within a cluster. Growth rates have been measured for seven strains to date and fall in the range of 0.3 to 0.8 hr⁻¹ (*Jemielita et al., 2014*; *Wiles et al., 2016*; *Schlomann et al., 2019*; *Wiles et al., 2020*); we use an intermediate value of 0.5 hr⁻¹ in all simulations below. In large systems, it is often valid to ignore fluctuations, in which case the model can be summarized by a single, deterministic equation for the likelihood of clusters of each size, for which analytic results are possible in some cases. In contrast, for small systems, which includes our experiments, random fluctuations will likely be relevant, and so we turn to computer simulations that capture stochastic dynamics.

We previously showed that a version of this model with all parameters measured (i.e., no remaining free parameters) generates a size distribution consistent with that of *Enterobacter* ZOR0014 (*Schlomann et al., 2019*). However, it was not clear which processes generated which features of

the distribution, or how generalizable the model was. Therefore, we studied this model in more detail, starting from a simplified version and iteratively adding complexity.

The observation that all distributions appeared to be organized around $P(\text{size}>n) \sim n^{-1}$ inspired us to consider connections to a classic populations genetics model that has this form for the distribution of allele frequencies, known as the Yule-Simons process (**Yule, 1925**; **Simon, 1955**; **Altan-Bonnet et al., 2020**; **Neher and Hallatschek, 2013**). An exponentially growing population subject to random neutral mutations that occur with probability $\epsilon$ will amass an allele frequency distribution that follows $P(\text{frequency}>x) \sim x^{-\frac{1}{1-\epsilon}}$ for large sizes, with the limit to $P(\text{frequency}>x) \sim x^{-1}$ for rare mutation. This heavy-tailed distribution reflects 'jackpot' events in which mutants that appear early rise to large frequencies through exponential growth. As long as mutation is rare compared to replication, this process robustly generates distributions $P(\text{size}>n) \sim n^{-1}$, without the need for fine tuning of the microscopic details. We therefore saw it as an attractive hypothesis for generating similar size distributions across diverse bacterial species.

Analogously, the size of mutant clones maps onto the size of a bacterial cluster, and the mutation process that generates new clones maps onto the fragmentation process that generates new clusters (**Figure 3A**). In situations where all cells in a cluster have the same probability of fragmenting, this analogy is exact and the same distribution emerges (Appendix). However, gut bacterial clusters are three-dimensional and likely encased in mucus (**van der Waaij et al., 1996**), so spatial structure likely influences fragmentation rates. We hypothesized that this spatial structure could be a mechanism for generating distributions shallower than $P(\text{size}>n) \sim n^{-1}$ that we observe in the data for large sizes (**Figure 2**) but that cannot be produced by the standard Yule-Simons mechanism. Therefore, we modified the Yule-Simons process by decoupling the growth and fragmentation processes and invoking a fragmentation rate, $F_n$, that scales as a power of the cluster size, $F_n \sim \beta n^{\nu_F}$ (**Figure 3B**). A value of $\nu_F = 1$ corresponds to the well-mixed limit of the Yule-Simons process. A value of $\nu_F = 2/3$ corresponds to only bacteria on the surface of clusters being able to fragment. An extreme value of $\nu_F = 0$ means that all clusters have the same rate of fragmenting, regardless of their size, and can be thought of as representing a chain of cells where only the cells at ends of the chain can break off.

In stochastic simulations of this model (Materials and methods) we find broad, power-law-like distributions for each value of $\nu_F$ (**Figure 3C**), but no signature of a shallow plateau at larger sizes. We define μ as the exponent of the probability distribution, $p(n) \sim n^{-\mu}$, such that the cumulative distribution function has the form $P(\text{size}>n) \sim n^{-\mu+1}$ (the latter is proportional to the integral of the former). Following established methods, we fit a power law, $P(\text{size}>n) \sim n^{-\mu+1}$ for $n>n_{\min}$ to simulation outputs using maximum likelihood estimation (**Clauset et al., 2009**) and examined the dependence on fragmentation rate. Faster fragmentation results in larger values of μ, reflecting steeper distributions, with the dependence being superlinear for $\nu_F = 1$, approximately linear for $\nu_F = 2/3$, and sublinear for $\nu_F = 0$ (**Figure 3D**, circles). Increasing values of $\nu_F$ also appeared to have increasing minimum values of μ, corresponding to rare fragmentation.

The minimum value of the distribution exponent can be computed by considering, for example, the total rate of fragmentation events. Denoting the total number of clusters by $M$ and the number of clusters of size $n$ by $c_n$, the rate of cluster production follows $\dot{M} \approx \beta \sum_n n^{\nu_F} c_n$ (Appendix). Assuming a power-law solution $c_n \sim n^{-\mu}$ and approximating the sum by an integral, we see that the rate of cluster production is finite only if

$$\mu > \nu_F + 1, \tag{1}$$

consistent with simulations. Therefore, spatial structure—modeled by decreasing $\nu_F$—is indeed a mechanism to generate distributions shallower than $P(\text{size}>n) \sim n^{-1}$. A heuristic argument for the rate dependence of the exponents in the long time, large size limit is provided in the Appendix, with the results summarized in **Table 2** and plotted as solid lines in **Figure 3D**. The analytic results agree reasonably well with simulations, with deviations becoming prominent as $\beta/r \approx 1$.

In summary, we identified a minimal growth-fragmentation process that generates power-law distributions with tuneable exponents in the experimentally observed range. However, this model does not include other features known to occur in the experimental system, including a finite carrying capacity that limits growth, cluster aggregation, and cluster expulsion, which may alter the asymptotic distributions. Moreover, this model fails to capture the large-size behavior of many of the

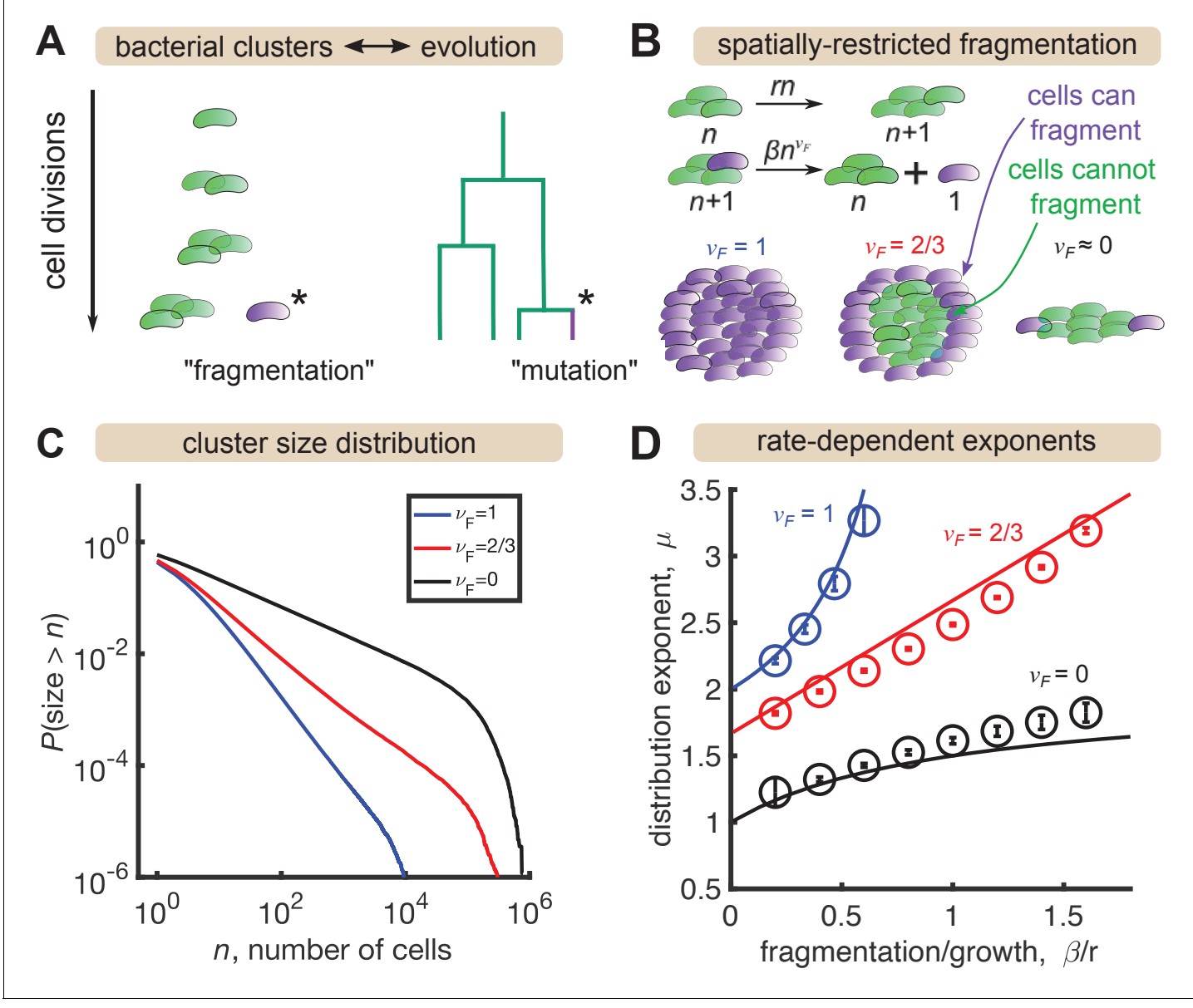

**Figure 3.** A minimal model inspired by evolutionary dynamics generates power law distributions. (A) Fragmentation is analogous to mutation and we can construct a genealogy that mirrors the physical structure of the clusters. (B) Summary of a growth/fragmentation process that includes the effect of spatially confined clusters. (C) Examples of reverse cumulative size distributions obtained from stochastic simulations of the model for different values of the fragmentation exponent, $\nu_F$. The tails of the distribution are approximately power laws, defined as $P(\text{size}>n) \sim n^{-\mu+1}$. Parameters: $r = 0.5$ hr$^{-1}$, $\beta = 0.4, 0.2, 0.167$ hr$^{-1}$, for $\nu_F = 0, 2/3, 1$, respectively, time $t = 24$ hr, and the system was initialized with 10 single cells. (D) Dependence of the resulting distribution exponent, $\mu$, on ratio of fragmentation to aggregation rate ($\beta/r$) and fragmentation exponent ($\nu_F$). Markers show mean and standard deviation across 100 simulations. Solid lines are approximate analytic results (*Table 2*). Parameters: same as (C) with $\beta$ varying.

The online version of this article includes the following source data for figure 3:

**Source data 1.** Results of power-law fits to simulated distributions.

experimental distributions, which exhibit a plateau (*Figure 2*). Therefore, we investigated extensions of the model.

**Table 2.** Analytic results for the minimal growth-fragmentation process.
Distribution exponent, μ, as a function of fragmentation exponent, $\nu_F$, fragmentation rate, β, and growth rate, r, as plotted in **Figure 3D**. Results are expected to be valid for long times ($t \to \infty$), large sizes ($n \to \infty$), and slow fragmentation ($\beta/r < 1$). See Appendix for details.

| | $\nu_F = 1$ | $\nu_F = 2/3$ | $\nu_F = 0$ |
|---|---|---|---|
| distribution exponent, μ | $1 + \frac{1}{1-\beta/r}$ | $\frac{5}{3} + \frac{\beta}{r}$ | $1 + \frac{\beta/r}{1+\beta/r}$ |

## Size-dependent aggregation enhances the abundance of large clusters

We explored a number of potential mechanisms for generating plateaus in the size distribution at large cluster sizes. As shown below, several plausible models fail to produce this feature. It emerges, however, from the incorporation of size-dependent aggregation rates.

First we tested whether finite time effects could introduce plateaus to the distributions of the minimal growth-fragmentation model, since our power-law solutions are only valid asymptotically. Indeed, stochastic simulations with $\nu_F = 1$ and rare fragmentation ($r = 0.5$ hr$^{-1}$, $\beta = 0.05$ hr$^{-1}$) showed that for systems initialized with 10 single cells (a reasonable comparison with initial colonization in the experiments *Wiles et al., 2016*), slight curvature appears in the distribution that weakens with time but is still detectable at 24 hr (*Figure 4—figure supplement 1*, circles). We confirmed that this effect was solely due to dynamics and not to any finite system effect by numerically integrating the master equation for this model, which describes the deterministic dynamics of an infinite system yet agrees with the stochastic simulation results (*Figure 4—figure supplement 1*, lines; Materials and methods). However, the curvature observed at finite times is substantially smaller than what occurs for some of the strains, such as *Enterobacter* ZOR0014 and *Vibrio* ZOR0036, so we believe it is not the dominant effect.

We next asked whether including additional processes to the model could produce the plateau effect, focusing on stationary distributions. As discussed above, populations in the larval zebrafish gut are known to reach carrying capacities that halt growth (*Jemielita et al., 2014*). Since we believe fragmentation is tied to growth, we modeled this as the fragmentation rate being slowed as the total number of cells, $N$, approaches carrying capacity, K, in the same way as the growth rate: $r \to r(1 - N/K)$ and $\beta \to \beta(1 - N/K)$. Carrying capacities have been estimated to range from $10^3$-$10^6$ cells, depending on the bacterial strain (*Jemielita et al., 2014*; *Wiles et al., 2016*; *Schlomann et al., 2018*; *Wiles et al., 2020*).

With this addition to the model, fragmentation halts in the steady state. However, in the larval zebrafish gut it has been well-documented that large bacterial aggregates are quasi-stochastically expelled out the intestine, after which exponential growth by the remaining cells is restarted (*Wiles et al., 2016*). We modeled expulsion by having clusters removed from the system altogether at a size-dependent rate $E_n = \lambda n^{\nu_E}$. It is unclear what value of the exponent $\nu_E$ best describes the experimental system, but previous studies measured expulsion rates for the largest clusters, typically of order $K \sim 10^3$ cells, in the range of 0.07 to 0.11 hr$^{-1}$ (*Wiles et al., 2016*; *Schlomann et al., 2019*). Therefore, we co-varied $\nu_E$ and λ such that $\lambda K^{\nu_E} \sim 10^{-1}$ hr$^{-1}$. Combining finite carrying capacity and expulsion leads to a non-trivial stationary distribution of the model that lacks a plateau for $\nu_E = 0$, 1/3, or 2/3 (*Figure 4—figure supplement 2*).

Finally, we considered the effect of cluster aggregation, which has been directly observed in live imaging experiments (*Schlomann et al., 2019*). We model aggregation with pairwise interactions where clusters come together and form a single cluster with size equal to the sum of the individual sizes. The aggregation rate is allowed to be size-dependent with the homogenous kernel $A_{nm} = \alpha(nm)^{\nu_A}$. As with expulsion, it is not clear which exponent value is the most realistic. Accurate measurements of aggregation rates are lacking, but we estimate bounds to be between 1 and 100 total aggregation events per hour for a typical population (Materials and methods), so we consider only pairs of $\alpha$ and $\nu_A$ that match these bounds. Further, an important theoretical distinction is that in purely aggregating systems, models with $\nu_A \geq 1/2$ exhibit a finite-time singularity corresponding to a gelation transition, at which point the distribution acquires a power-law tail, while distributions have exponential tails when $\nu_A < 1/2$ (*Krapivsky et al., 2010*). We considered both regimes.

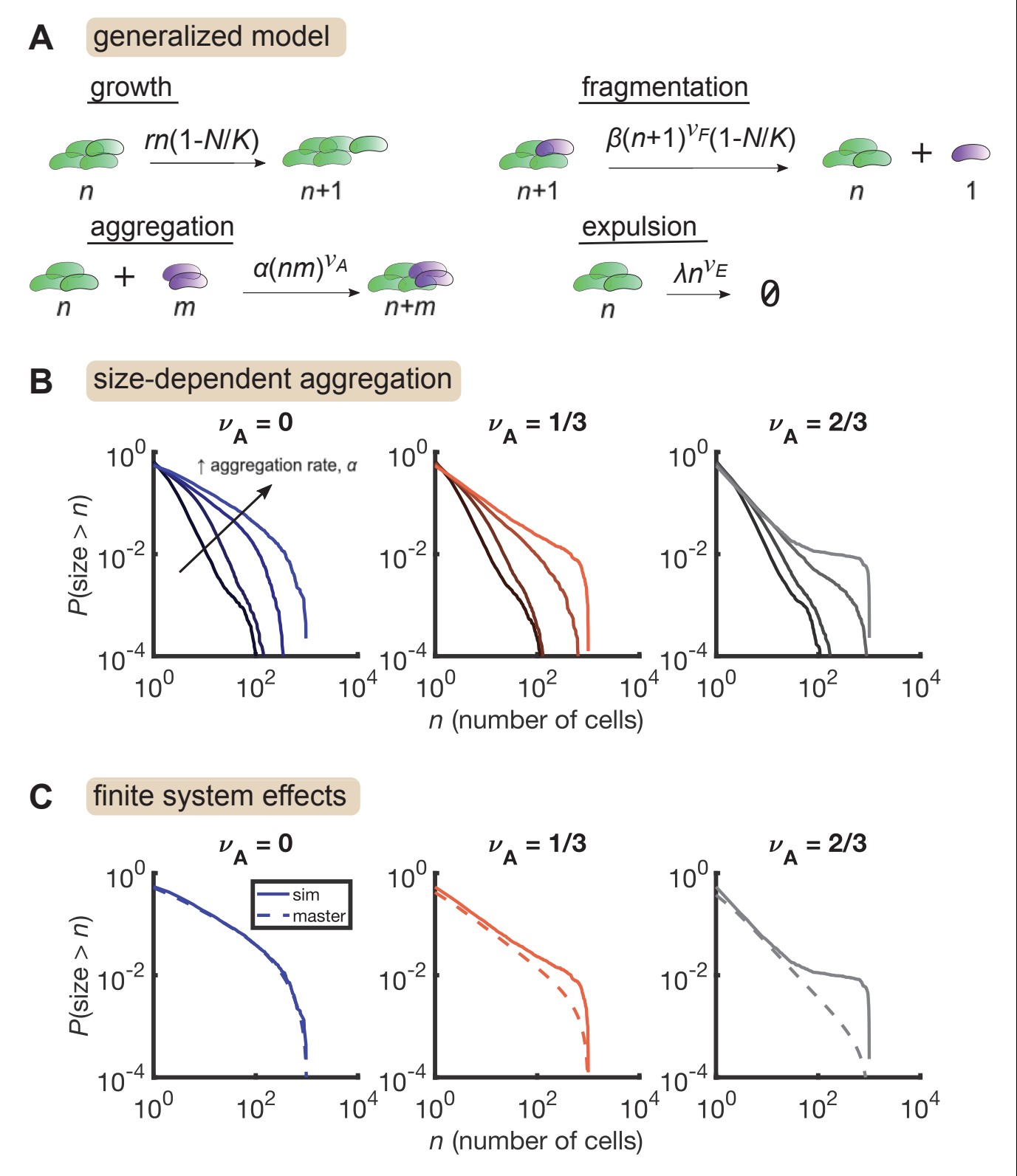

**Figure 4.** Size-dependent aggregation introduces a plateau in the size distribution. (A) Schematic of the generalized model. Parameters summarized in *Table 3*. (B) Reverse cumulative distributions obtained from simulations for different values of $\nu_A$ (left, middle, right) and $\alpha$ (different colored lines within each panel). Increasing aggregation produces a plateau if the aggregation depends strongly enough on cluster size. (C) The plateau arises only in stochastic simulation of finite systems with size-dependent aggregation. Solid lines are stochastic simulations, dashed lines are the result of numerically

*Figure 4 continued*

integrating the master equation. Parameters: $r = 0.5$ hr$^{-1}$, $\nu_F = 2/3$, $\beta = 0.5$ hr$^{-1}$, $\nu_E = 1/3$, $\lambda = 0.01$ hr$^{-1}$, $K = 10^3$, and the number of simulation was replicates = 150 per parameter set. For each value of $\nu_A$, we considered $\alpha$ values of 0 (no aggregation) and then varied $\alpha$ logarithmically, with the following (min, max) values for $\log_{10} \alpha$: (−4,−2) for $\nu_A = 0$, (−4.5,−2.5) for $\nu_A = 1/3$, and (−5,−3) for $\nu_A = 2/3$.

The online version of this article includes the following figure supplement(s) for figure 4:

**Figure supplement 1.** Distributions of growth/fragmentation process at short times.

**Figure supplement 2.** Distributions for a process with density-dependent growth, fragmentation, and expulsion.

**Figure supplement 3.** Distributions for a model with only aggregation and $\nu_A = 1$.

We added aggregation to our growth-driven process and arrived at the general model described in *Figure 4A*. Parameters are also summarized in *Table 3*. Strikingly, we found that increasing aggregation rate produces the large-size plateau seen in our data, but only when aggregation rate scales sufficiently quickly with cluster size (*Figure 4B*, right, $\nu_A = 2/3$) and not when aggregation is size-independent (*Figure 4B*, left, $\nu_A = 0$). A mild effect is observed for $\nu_A = 1/3$ (*Figure 4B*, middle). The largest plateau (*Figure 4B*, $\nu_A = 2/3$, highest curve) corresponds to $15 \pm 3$ (mean ± std. dev) total aggregation events per hour. This value is consistent with our rough experimental bounds of 1–100 hr$^{-1}$.

We further found that this plateau effect is intrinsic to finite systems (*Figure 4C*). For the most aggregated cases in *Figure 4B*, we numerically solved the corresponding master equation, representing the deterministic dynamics of an infinite system, and found that the plateau did not occur. Master equation and stochastic simulation solutions agree for $\nu_A = 0$, but for larger values of $\nu_A$, the two solutions only agree in the small size regime. At large sizes, stochastic simulations produce an overabundance of large clusters compared to the master equation solution. This result indicates that in a finite system, strong aggregation can deplete small clusters, condensing them into a small number of large clusters on the order of the system-size.

This overall process is reminiscent of the gelation transition in soft materials. Stochastic dynamics of finite systems of purely aggregating particles at the gelation transition also produces distributions with plateaus, but with an initial decay given approximately by a power law with $\mu = 5/2$ (*Figure 4— figure supplement 3*, see also *Matsoukas, 2015*). Combined with a growth/fragmentation/expulsion process, we found that size-dependent aggregation produces a distribution that initially decays in a power-law-like manner with tunable exponents and then exhibits a tuneable plateau, as we observe in the experimental data.

**Table 3.** Summary of model variables and parameters.

| Variable/parameter | Description |
| --- | --- |
| $n$ | Cluster size (number of cells) |
| $p(n)$ | Probability of cluster size, $n$ |
| $P(\text{size} > n)$ | Cumulative probability; probability of size being larger than $n$ |
| $\mu$ | Exponent of power law; $p(n) \sim n^{-\mu}$, $P(\text{size} > n) \sim n^{-\mu+1}$ |
| $r$ | Cell division rate |
| $K$ | Carrying capacity; maximum number of cells |
| $\beta$ | Fragmentation rate |
| $\nu_F$ | Fragmentation exponent; clusters of size $n$ fragment with rate $\beta n^{\nu_F}$ |
| $\alpha$ | Aggregation rate |
| $\nu_A$ | Aggregation exponent; clusters of sizes $n$ and $m$ aggregate with rate $\alpha (nm)^{\nu_A}$ |
| $\lambda$ | Expulsion rate |
| $\nu_E$ | Expulsion exponent; clusters of size $n$ are expelled with rate $\lambda n^{\nu_E}$ |

## Discussion

We analyzed image-derived measurements of bacterial cluster sizes from larval zebrafish intestines and discovered a common family of size distributions shared across bacterial species. These distributions are extremely broad, exhibiting a power-law-like decay at small sizes that becomes shallower at large sizes in a strain-specific manner. We then demonstrated how these distributions emerge naturally from realistic kinetics: growth and single-cell fragmentation together generate power-law distributions, analogous to the distribution of neutral alleles in expanding populations, while size-dependent aggregation leads to a plateau representing the depletion of mid-sized clusters in favor for a single large one. In summary, we found that gut bacterial clusters are well-described by a model that combines the features of evolutionary dynamics in growing populations with those of inanimate systems of aggregating particles; intestinal bacteria form a 'living gel'.

Gels are characterized by the emergence of a massive connected cluster that is on the order of the system size. In the larval zebrafish intestine, we often find for some bacterial species that the majority of the cells in the gut are contained within a single cluster, similar to a gel-like state. While growth by cell division generates large clusters, it is the aggregation process that leads to system-sized clusters being over-represented. This enhancement of massive clusters manifests as a plateau in the size distribution and is reminiscent of a true gelation phase transition. In our model, the prominence of this plateau appears to follow the same trend as in non-living, purely aggregating systems: the plateau depends strongly on the aggregation exponent that dictates the size-dependence of aggregation, with exponents larger than $1/2$ leading to strong plateaus and exponents less than $1/2$ leading to weak or no plateaus.

How this strong size-dependence in aggregation emerges in the intestine is unclear, although we hypothesize that active mixing by intestinal contractions, which can in fact merge multiple clusters at once (*Schlomann et al., 2019*), is an important driver. We envision that the exponents for aggregation and also for fragmentation are likely generic, set by physical aspects of the intestine and the geometry of clusters, while the rates of these processes are bacterial-species dependent. In our system, we predict that differences in aggregation and/or fragmentation rates between strains underly the differences in measured size distributions. Further, it is possible that individual bacteria can tune these rates by altering their behavior, for example, modulating swimming motility (*Wiles et al., 2020*), in response to environmental cues. Quantitatively understanding how the combination of intestinal fluid mechanics and bacterial behaviors determine aggregation and fragmentation rates would be a fruitful avenue of future research. More abstractly, active growth combined with different aggregation processes, for example the fractal structures of diffusion-limited aggregation, may lead to different families of size distributions that would be interesting to explore.

On the experimental front, direct measurements of aggregation and fragmentation rates from time-lapse imaging would be an extremely useful next step. However, these measurements are technically challenging. Even by eye, unambiguously identifying that a single bacterium fragmented out of a larger aggregate, and did not simply float into the field of view, requires faster imaging speeds than we can currently obtain. Sparse, two-color labeling may improve reliability of detection, but would decrease the frequency of observing an event. Automatic identification of fragmentation events in time-lapse movies is a daunting task, but recent computational advances, for example using convolutional neural networks to automatically identify cell division events in mouse embryos (*McDole et al., 2018*), may provide a good starting point. Aggregation is easier to observe by eye, but its automatic identification presents similar challenges in analysis.

Given the general and minimal nature of the model's assumptions, we predict that the form of the cluster size distributions we described here is common to the intestines of animals, including humans. This prediction of generality could be tested in a variety of systems using existing methods. In fruit flies, live imaging protocols have been developed that have revealed the presence of three-dimensional gut bacterial clusters highly reminiscent of what we observe in zebrafish, particularly in the midgut (*Koyama et al., 2020*). Quantifying the sizes of these clusters would allow further tests of our model.

In mice, substantial progress has been made in imaging histological slices of the intestine with the luminal contents preserved (*Tropini et al., 2017*). Intestinal contents are very dense in the distal mouse colon, however, and it is not clear how one should define cluster size. Other intestinal regions are likely more amenable to cluster analysis. Moreover, with species-specific labeling, it is possible

to measure the distribution of clonal regions in these dense areas (*Mark Welch et al., 2017*). One could imagine then comparing these data to a spatially-explicit, multispecies extension of the model we studied here.

Our model could also be tested indirectly for humans and other animals incompatible with direct imaging by way of fecal samples. Two decades ago, bacterial clusters spanning three orders of magnitude in volume were observed in gently dissociated fecal samples stained for mucus, but precise quantification of size statistics was not reported (*van der Waaij et al., 1996*). Repeating these measurements with quantification, from for example imaging or flow cytometry, would also provide a test of our model, albeit on the microbiome as a whole rather than a single species at a time. The interpretation therefore would be of an effective species with kinetic rates representing average rates of different species.

To close, we emphasize that the degree of bacterial clustering in the gut is an important parameter for both microbial population dynamics and host-bacteria interactions. More aggregation leads to larger fluctuations in abundance due to the expulsion of big clusters, and also thereby increase the likelihood of extinction (*Schlomann et al., 2019*; *Schlomann and moments, 2018*). Further, aggregation within the intestinal lumen can reduce access to the epithelium and reduce pro-inflammatory signaling (*Wiles et al., 2020*). Therefore, measurements of cluster sizes may be an important biomarker for microbiota-related health issues, and inference of dynamics from size statistics using models like this one could aid the development of therapeutics.

## Materials and methods

### Key resources table

| Reagent type (species) or resource | Designation | Source or reference | Identifiers | Additional information |
|---|---|---|---|---|
| Software, algorithm | Analysis code | This study | | see Materials and methods, Simulations |
| Other | Cluster size data | *Schlomann and moments, 2018* | | |
| Other | Cluster size data | *Schlomann et al., 2019* | | |
| Other | Cluster size data | *Wiles et al., 2020* | | |

### Data

We assembled data on gut bacterial cluster sizes from three different studies on larval zebrafish (*Schlomann et al., 2018*; *Wiles et al., 2020*). Size data from *Schlomann et al., 2018* and *Schlomann et al., 2019* were taken directly from the supplementary data files associated with those publications. The raw size data from *Wiles et al., 2020* was not included in its associated supplementary data file, but summary statistics such as planktonic fraction were. All sizes were rounded up to the nearest integer.

Details of experimental procedures can be found in the original papers. In brief, as described in *Figure 1*, animals were reared germ-free, mono-associated with a single bacterial strain, each carrying a chromosomal GFP tag, and then imaged 24 hr later using a custom-built light sheet fluorescence microscope (*Jemielita et al., 2014*). The gut is imaged in four tiled sub-regions that are registered via cross-correlation and manual adjustment. Imaging a full gut volume ($\approx 1200$ µm $\times$ 300 µm $\times$ 150 µm) with 1 µm slices takes approximately 45 s. Laser power (5 mW) and exposure time (30 ms) were identical for all experiments.

The image analysis pipeline used to enumerate bacterial cluster sizes is also described in detail in the original publications and in reference (*Jemielita et al., 2014*). In brief, single cells (small objects) and multicellular aggregates (large objects) are identified separately. The number of cells per aggregate is then estimated as the total fluorescence intensity of the aggregate divided by the mean fluorescence intensity of a single cell. Small objects are identified in three dimensions with a combination of difference-of-gaussians and wavelet filters (*Olivo-Marin, 2002*) and then culled using a support vector machine classifier and manual curation. Large objects are segmented in maximum intensity projections using a graph-cut algorithm (*Boykov and Kolmogorov, 2004*) seeded by either

an intensity- or gradient-thresholded mask. The total intensity of an aggregate is computed by extending the two-dimensional mask in the $z$-direction and summing fluorescence intensities above a threshold calculated from the boundary of the mask, with pixels detected as part of single cells removed. The boundary of the gut is manually outlined prior to image analysis and used to exclude extra-intestinal fluorescence.

## Size distribution

For the experimental data, reverse cumulative distributions were computed as

$$P(\text{size>n}) = \frac{\text{number of clusters with size>n}}{\text{total number of clusters}}. \tag{2}$$

In combining data from different samples colonized with the same strain, we pooled together all sizes and computed the distribution in the same way. For simulations with large numbers clusters, we computed this distribution iteratively, looping through each simulation replicate and independently updating (number clusters with size $>n$) and (total number of clusters), and normalizing at the end.

For the binned probability densities in *Figure 2—figure supplement 1*, data were similarly pooled across samples and then sorted into logarithmically spaced bins of $log_{10}$ width = 0.4.

## Estimates on bounds of agg rates

We estimated approximate bounds on the rate of total aggregation events as follows. For the maximum rate, we note that a typical population contains approximately 200 clusters (mean ± std. dev of 244 ± 182). In the absence of other processes, condensing this system into one cluster would require 100 aggregation events. Populations consisting of almost entirely one large cluster are rare but have been documented (*Schlomann et al., 2018*). Therefore, we estimate that this complete condensation can occur no more than once an hour, leading to an upper bound on the total rate of aggregation events of 100 per hour.

For the minimum rate, we start with the observation that aggregation has been directly observed between small clusters and also between small clusters and a single large cluster during a large expulsion event (*Schlomann et al., 2019*). Considering just the latter process, we know that large expulsion events happen roughly once every 10 hr. If approximately 10 small clusters are grouped into the large cluster during transit out of the gut, that would correspond 10 total aggregation events in 10 hr, or, 1 per hour, which we take as a lower bound.

## Simulations

We used three different numerical approaches for studying the models discussed here. The minimal growth-fragmentation process in *Figure 3* was simulated with a Poisson tau-leaping algorithm *Gillespie, 2001* with a simple fixed tau value of $\tau = 0.1$ hr. At each time step, the number of growth and fragmentation events was drawn from a Poisson distribution with the rates given in *Figure 3B* along with the constraint that clusters must be of size two or larger to fragment.

For the full model including aggregation and expulsion, we used Gillespie's algorithm *Gillespie, 1977* for fragmentation, aggregation, and expulsion events, while growth was updated deterministically according to a continuous logistic growth law approximated by an Euler step with $dt = \min(\tau, 0.1\text{hr})$, where $\tau$ here refers to the time to next reaction. For the Gillespie steps, if the time to next reaction exceeded the doubling time, $(\ln 2)/r$, the growth steps were performed and then the propensity functions were re-calculated.

Finally, we compared these stochastic simulations to a model in the thermodynamic limit where individual clusters are replaced with cluster densities that evolve deterministically, which is referred to as a master equation (*Krapivsky et al., 2010*). The master equation for the general model reads

$$\dot{c}_n = \frac{\alpha}{2} \sum_{m=1}^{n} [(n-m)(m)]^{\nu_A} c_{n-m} c_m - \alpha n^{\nu_A} c_n \sum_m m^{\nu_A} c_m$$
$$+ r\left(1 - \frac{N}{K}\right)[(n-1)c_{n-1} - nc_n] - \lambda n^{\nu_E} c_n$$
$$+ \beta\left(1 - \frac{N}{K}\right)\left((n+1)^{\nu_F} c_{n+1} - n^{\nu_F} c_n + \delta_{n,1} \sum_m m^{\nu_F} c_m\right).$$

This set of equations was solved numerically on a bounded size grid using an Euler method with step size $dt = 0.0001$ hr. Models that include a carrying capacity, $K$, are already defined on a finite domain of integers ranging from one to $K$ and the master equation is naturally represented by a set of $K$ ordinary differential equations. For models without a carrying capacity, we introduced a maximum size given by the average population size at the last time point, $n_{max} = \exp(rt_{max})$ (rounded up to the nearest integer), and used reflecting boundary conditions at $n_{max}$.

A distribution was deemed stationary if it was visibly unchanged after an additional 50% of simulation time.

MATLAB code for simulating these models and plotting data can be found at https://github.com/rplab/cluster_kinetics (copy archived at swh:1:rev:f55a54a9c88e4fb8376dfc91e25ac4383c4240ae, *Schlomann, 2021*).

### Estimating distribution exponents

For the simulated distributions in *Figure 3* we estimated a power law exponent using the maximum likelihood-based method described in *Clauset et al., 2009* and the plfit.m code supplied therein. This model includes a minimum size as a free parameter that dictates when the power-law tail begins. The minimum size is chosen to minimize the Kolmogorov-Smirnov distance between the data and model distributions for sizes greater than the minimum size. Best fit values of the exponent and minimum size are included in *Figure 3—source data 1*.

For the experimentally measured distributions, we used both maximum likelihood estimation and linear fitting to the log-transformed cumulative distribution to calculate exponents.

## Acknowledgements

We thank Jayson Paulose for helpful discussions. Research was supported by the National Institutes of Health (http://www.nih.gov/), under Awards P50GM09891, P01GM125576, F32AI112094, and T32GM007759. Work was also supported by the National Science Foundation under Award 1427957, and an award from the Kavli Microbiome Ideas Challenge, a project led by the American Society for Microbiology in partnership with the American Chemical Society and the American Physical Society and supported by The Kavli Foundation. The University of Oregon Zebrafish Facility is supported by a grant from the National Institute of Child Health and Human Development (P01HD22486). BHS was supported by the James S McDonnell Foundation postdoctoral fellowship. The funders had no role in study design, data collection and analysis, decision to publish, or preparation of the manuscript.

## Additional information

### Funding

| Funder | Grant reference number | Author |
| --- | --- | --- |
| National Institutes of Health | P50GM09891 | Brandon H Schlomann Raghuveer Parthasarathy |
| National Institutes of Health | P01GM125576 | Brandon H Schlomann Raghuveer Parthasarathy |
| National Institutes of Health | F32AI112094 | Brandon H Schlomann Raghuveer Parthasarathy |
| National Institutes of Health | T32GM007759 | Raghuveer Parthasarathy |
| National Science Foundation | 1427957 | Brandon H Schlomann Raghuveer Parthasarathy |
| James S. McDonnell Foundation | | Brandon H Schlomann |
| Kavli Foundation | Kavli Microbiome Ideas Challenge | Brandon H Schlomann Raghuveer Parthasarathy |
| National Institutes of Health | P01HD22486 | Brandon H Schlomann Raghuveer Parthasarathy |

The funders had no role in study design, data collection and interpretation, or the decision to submit the work for publication.

## Author contributions
Brandon H Schlomann, Raghuveer Parthasarathy, Conceptualization, Resources, Software, Formal analysis, Supervision, Funding acquisition, Investigation, Methodology, Writing - original draft, Writing - review and editing

## Author ORCIDs
Brandon H Schlomann ⓘD https://orcid.org/0000-0003-2280-0132
Raghuveer Parthasarathy ⓘD https://orcid.org/0000-0002-6006-4749

## Ethics
Animal experimentation: The studies that generated the data analyzed in this paper (see cited references) were done in strict accordance with protocols approved by the University of Oregon Institutional Animal Care and Use Committee and following standard protocols.

## Decision letter and Author response
Decision letter https://doi.org/10.7554/eLife.71105.sa1
Author response https://doi.org/10.7554/eLife.71105.sa2

# Additional files
## Supplementary files
• Supplementary file 1. Cumulative distribution exponents of cluster size distributions by strain, with analysis of sensitivity to single-cell detection. The small size regime of the cluster size distributions were fit to a power-law model for sizes up to 100 cells using two methods: a linear fit to $P(\text{size}>n)$ and maximum likelihood estimation (Materials and methods). The fits were done for each animal and the resulting mean ± std. dev of the exponents (corresponding to $\mu - 1$, as defined in the text) are given for each strain. For each method, the fits were done twice, once including single cells, and once considering only cells of size two or greater. As discussed in the main text, the largest uncertainty in cluster size enumeration from the images occurs at small sizes. Ignoring single cells in the fit only mildly changes the average exponent, and all changes are within uncertainties. Most exponents are consistent with $\mu - 1 = 1$.

• Transparent reporting form

## Data availability
A table of all bacterial cluster sizes analysed in this study is included in the Source Data Files. MATLAB code for simulating the models described in the study is available at https://github.com/rplab/cluster_kinetics (copy archived at https://archive.softwareheritage.org/swh:1:rev:f55a54a9c88e4fb8376dfc91e25ac4383c4240ae).

The following datasets were generated:

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

## Appendix 1

### Analytic calculations for growth-fragmentation processes

We consider a model with only growth and fragmentation processes and make heuristic arguments for the form of the asymptotic size distribution. In particular, we are interested in how the exponent of the resulting power law tails depends on the growth and fragmentation rates. We derive here the results listed in *Table 2* of the main text.

### Model summary

Clusters grow according to

$$n \rightarrow n+1 \text{ with rate } rn, \tag{3}$$

and they fragment according to

$$n+1 \rightarrow n \text{ with rate } \beta n^{\nu_F} \text{ for } n>1. \tag{4}$$

The cell lost during fragmentation becomes its own cluster of size one.

We now consider the deterministic dynamics of a large system. Putting these reactions together, we can write the master equation for the density of cells of size $n$, $c_n$:

$$\dot{c}_n = \beta \left( (n+1)^{\nu_F} c_{n+1} - n^{\nu_F} c_n + \delta_{n,1} \sum_m m c_m \right) + r((n-1)c_{n-1} - nc_n). \tag{5}$$

In what follows we will use the terms ''density'' and ''total number'' interchangeably, measuring volume in units of our system size (i.e., number of cells per gut). The first moment of this equation gives the total number of cells,

$$\dot{N} = rN. \tag{6}$$

The zeroth moment gives the total number of clusters,

$$\dot{M} = \beta \left( \sum_{n=1}^{\infty} n^{\nu_F} c_n - c_1 \right). \tag{7}$$

Here, the $c_1$ term reflects the fact that in this model, cells must have size 2 or greater to fragment.

Finally, in a continuum picture, the size of a particular cluster is described by

$$\dot{n} = rn - \beta n^{\nu_F} \tag{8}$$

A well-known heuristic derivation of the stationary distribution of this type of process is based on the relationship between the number of clusters, $M = \sum_n c_n$, and the total number of cells, $N = \sum_n n c_n$. The key to this derivation is to recognize that $M(t)$ acts as a proxy for the rank of the cluster that arises at time $t$: for the $j^{th}$ cluster to arise, there are $j-1$ clusters that have a larger size, since the relative ordering of cluster sizes is preserved during exponential growth. For large sizes, when cluster rank is expressed as a function of cluster size it becomes proportional to the reverse cumulative distribution function, from which we obtain the density.

It turns out that the differences in behaviors of exponents measured in simulations for different values of $\nu_F$ can be understood by considering the importance of two terms in particular: the $c_1$ term in the equation for $M$, and the $\beta n^{\nu_F}$ term in the equation for $n$.

Case 1: $\nu_F = 1$

The total number of cells follows simple exponential growth, $N(t) \sim \exp(rt)$. For $\nu_F = 1$, the total number of clusters is governed by the equation

$$\dot{M} = \beta(N - c_1), \tag{9}$$

where the $c_1$ term arises in our model because clusters can only fragment if they have size $\geq 2$. At

long times, however, we expect $c_1 \ll N$ and we therefore ignore this term, leading to $M(t) \sim (\beta/r)\exp(rt) \sim N(t)$. A cluster that arises at time $t'$ will at a later time $t$ have a size

$$n(t,t') = e^{(r-\beta)(t-t')}. \tag{10}$$

Ignoring overall $t$ dependence, we can express this size as a function of the rank of this cluster,

$$n \sim M^{-(1-\beta/r)}. \tag{11}$$

Inverting this relationship, and invoking the proportionality between $M$ and the reverse cumulative distribution, $P(\text{size}>n)$, results in

$$P(\text{size}>n) \sim M \sim n^{-\frac{1}{1-\beta/r}}, \tag{12}$$

and differentiating produces the expected result

$$c_n \sim n^{-\left(1+\frac{1}{1-\beta/r}\right)}, \tag{13}$$

where $c_n$ is normalized by $\sum c_n = M$. This result matches the traditional Yule-Simons process, where each organism divides at rate $r$ and then mutates with probability $\epsilon$, with $\epsilon = \beta/r$.

Case 2: $0 < \nu_F < 1$

In this case, when considering the equation for the size of a particular cluster,

$$\dot{n} = rn - \beta n^{\nu_F}, \tag{14}$$

for $\nu_F < 1$ we ignore the second term on the right hand side. This term represents loss due to fragmentation, can be ignored for large sizes. Specifically, we consider sizes greater than a critical size, $n_c = (\beta/r)^{1/(1-\nu_F)}$, below which clusters will shrink. Ignoring this term, the size of a particular clusters that arose at time $t'$ is given by

$$n(t) = e^{r(t-t')}. \tag{15}$$

The total number of clusters follows

$$\dot{M} = \beta\left(\sum_{n=1}^{\infty} n^{\nu_F} c_n - c_1\right). \tag{16}$$

Like above with $\nu_F = 1$, we ignore the $c_1$ term. Unlike for $\nu_F = 1$, we don't have a closed equation for the fractional moment $\sum_{n=1}^{\infty} n^{\nu_F} c_n$. Therefore, we take the approach of making a power law ansatz

$$c_n \equiv \frac{1}{Z} n^{-\mu} \tag{17}$$

with normalization

$$\frac{1}{Z}\sum_n n^{-\mu} = M, \tag{18}$$

and then solve for the exponent $\mu$ self-consitently First, we approximate the sums by integrals and arrive at an equation for $M$

$$\dot{M} = \beta\left(\frac{\mu-1}{\mu-\nu_F-1}\right)M. \tag{19}$$

Then we follow the same logic as for the $\nu_F = 1$ case. Solving for $M(t)$, we get

$$M(t) \sim \exp\left[\beta\left(\frac{\mu-1}{\mu-\nu_F-1}\right)t\right]. \tag{20}$$

Combining terms into

$$\eta(\mu) \equiv \beta\left(\frac{\mu-1}{\mu-\nu_F-1}\right), \tag{21}$$

we then relate the size of a cluster that arose at time $t'$ to the rank of that cluster, $M(t')$,

$$n \sim M^{-\frac{r}{\eta}} \tag{22}$$

from which we compute the scaling behavior of $c_n$,

$$c_n \sim n^{-\left(\frac{\eta}{r}+1\right)}. \tag{23}$$

Equating this expression with the original ansatz, we arrive at a self-consistency equation for $\mu$

$$\mu = \frac{\beta}{r}\left(\frac{\mu-1}{\mu-\nu_F-1}\right)+1, \tag{24}$$

which we solve to obtain an exponent linear in the rates,

$$\mu = \nu_F + 1 + \frac{\beta}{r}. \tag{25}$$

This result is plotted in **Figure 3D** of the main text with $\nu_F = 2/3$ and agrees reasonably well with simulations.

Case 3: $\nu_F = 0$

For $\nu_F = 0$, the equation for $M$ simplifies to

$$\dot{M} = \beta(M - c_1). \tag{26}$$

The simulation results in **Figure 3D** indicate that the relationship between $\mu$ and $\beta/r$ is no longer linear, which we expect to be due to the $c_1$ term reducing the propensity for fragmentation. That this single-cell effect is relevant for $\nu_F = 0$ makes sense because we expect most clusters to be of order 1, which would make $c_1$ of order $M$. To account for this term explicitly, we make the same power law ansatz as before,

$$c_n \equiv M(\mu-1)n^{-\mu} \tag{27}$$

and then extrapolate down to $n = 1$ to estimate $c_1$,

$$c_1 = M(\mu-1). \tag{28}$$

This extrapolation is purely a convenient approximation, as the distribution is likely not a true power law down to sizes of $\mathcal{O}(1)$. With this ansatz, the equation for $M$ reads

$$\dot{M} = \beta(\mu-1)\left(\frac{1}{\mu-1}-1\right)M. \tag{29}$$

Combining terms we can define

$$\eta \equiv \beta(\mu-1)\left(\frac{1}{\mu-1}-1\right) \tag{30}$$

and write

$$\dot{M} \equiv \eta(\mu)M. \tag{31}$$

Then, following the same protocol as above, we can relate the frequency of a cluster to its rank,

$$n \sim M^{-\frac{r}{\eta}} \tag{32}$$

from which we compute the scaling behavior of $c_n$,

$$c_n \sim n^{-\left(\frac{\eta}{r}+1\right)}. \tag{33}$$

Equating this result with the original ansatz, we get the self-consistency equation

$$\mu = \frac{\beta}{r}(\mu - 1)\left(\frac{1}{\mu - 1} - 1\right) + 1. \tag{34}$$

Solving this equation we get

$$\mu = 1 + \frac{\beta/r}{1 + \beta/r}. \tag{35}$$

This result is plotted in *Figure 3D* of the main text and agrees reasonably well with simulations, with notable deviations occurring once $\beta/r \approx 1$.

## Discussion

In this model growth and fragmentation are treated as separate processes. This choice is convenient in the context of the full model including aggregation because classic reversible aggregation models (*Krapivsky et al., 2010*) are contained within this general framework when growth and expulsion rates are set to zero, and also because fragmentation conserves total cell number. As a consequence of this choice, single cells are forbidden from fragmenting and don't contribute to the total rate of fragmentation events. This feature differs from common evolutionary variants of this model for asexual populations, where growth and mutation are linked. In those models, all organisms divide and in each division have a probability of mutating (analogous to fragmenting in our model), so the rate of mutant production scales with the total population, including singletons. This is mostly a minor difference, but, as we showed, it does lead to different behaviors of the resulting distribution exponents in the limit of fast fragmentation.

We showed that we can account for this effect when it is important (in the case $\nu_F = 0$), but we cannot say *when* it will be important. That is because this effect depends on the number of single cells, which lies outside the regime of our large-size asymptotics that underly the continuum approximation. Ultimately the effect depends on how large the the number of single cells, $c_1$, is compared to the fractional moment $\sum_n n^{\nu_F} c_n$. If the distribution has a significant shoulder, than $c_1$ may be smaller than the extrapolation of the power-law form down to $n = 1$. In that case, the single-cell effect may be less important than this extrapolation would predict it to be.

