## [Decision Letter]

**Acceptance summary:**

This study investigates bacterial aggregates in the larval zebrafish gut, revealing that the cluster distributions observed in vivo share mathematical similarities with those that occur during gel formation in soft condensed matter physics. The work links complex in vivo dynamics with a simple and potentially general biophysical phenomenon – making it an example of a "living" material in a natural biological context – and opens the door to new studies on the interplay between cell clustering and microbiome dynamics in a living host.

**Decision letter after peer review:**

Thank you for submitting your article "Gut bacterial aggregates as living gels" for consideration by *eLife*. Your article has been reviewed by 3 peer reviewers, and the evaluation has been overseen by a Reviewing Editor and Wendy Garrett as the Senior Editor. The following individuals involved in review of your submission have agreed to reveal their identity: William B Ludington (Reviewer #1); Srividya Iyer-Biswas (Reviewer #3).

Essential revisions:

The reviewers and editors appreciate the authors' innovative study on the physical basis for spatial aggregation in gut microbial communities, a timely topic that bridges soft condensed matter physics and microbiology. The reviewers noted many strengths of the work, including the elegant combination of model system and a simple biophysical model, the mechanistic insight provided by the model, the compelling agreement between data and theoretical predictions, and the potential generality of the proposed mechanism to other systems. We also commend the authors on code that is well documented, compact, and easily accessible.

We also identified several places where additional discussion would improve clarity and help the message reach a broader audience.

1) We would like to see an expanded explanation of the theoretical model to make the paper more accessible to a broader audience. For example, readers with a more biological background may find a more detailed discussion of the intuitive and mechanistic basis of the model helpful (see, in particular, the suggestions from Reviewer 2, including suggestions about adding context related to aggregation in other biological systems). Similarly, readers with a more physics background might find it helpful to connect the model (and the phenomenon) more directly to previous work – for example, the authors might note that this is a variant of a preferential attachment (Yule-Simons) model, and the authors could clarify the motivation for the particular model (e.g. what is special about the slope -1?). This discussion should also include further explanation of the gel characterization, and perhaps a brief introduction to gels in the Intro section, particularly since gel is mentioned in the title and abstract.

2) While not required, a sensitivity analysis (or similar discussion) that addresses potential sources of error in the parameter estimates would add to the work. How robust are the results to (for example) experimental errors in the measurements? In the absence of a detailed analysis, the authors could explicitly discuss any limitations imposed on the results due to these uncertainties.

3) The reviewers point out several places where the text could be strengthened and clarified, and also places where a general reader might benefit from a brief mention of future extensions or future work.

*Reviewer #1 (Recommendations for the authors):*

I really enjoyed reading this paper, and I learned from it. My comments below are meant to help improve the work.

1. Figure 1C: do the smaller clusters tend to be closer to the larger cluster? Why? Is that due to the interaction with fluid flows?

2. I think readers might find it helpful if the authors could note that this is a variant of a preferential attachment model, which I believe is better known than the Yule-Simons name.

3. Because single gut dynamic data is available, I would have appreciated seeing some time lapse imaging of clusters merging. Are fragmentations of large clusters into smaller clusters observed?

4. I found the use of the term fragmentation somewhat confusing. To me what is described with single cells falling off the cluster is more akin to dispersal of propagules. Is there actual fragmentation of large clusters into medium clusters? Or is the vast majority of fragmentation due to single cells? In that case, a Leslie matrix approach might be more appropriate. If the authors could clarify whether the fragmentation of a big cluster into medium clusters occurs, that would solve the point of confusion for me. If this type of fragmentation does not occur, fragmentation might not be the best term.

5. To make the paper align better with the title, I would have found it helpful if there had been some discussion of the importance of gels in the introduction. An argument could be that it is too mundane, but if such a simple and well understood model can be mechanistically applied to a complex system such as the gut microbiome, I would argue that is a major advance. The mechanistic cases presented in the supplement do a nice job of establishing the role of individual parameters in shaping the distribution.

6. I struggled a bit with the fact that expulsion was not considered until later in the manuscript. If the major factors examined (including expulsion) were mentioned in the last paragraph of the introduction, it would have helped me be patient in reading.

*Reviewer #2 (Recommendations for the authors):*

1. Results section: a more detailed explanation in the main text of how these processes are modeled by the theory would provide added support to the generalizability of the model and enable others to clearly see how they might consider test predictions in their own systems. For example, the description of the biological interpretation and limits of the distribution exponent µ would be very valuable.

2. Results section: How robust are the size distributions and model to potential error in bacterial cell enumeration? The number of cells per aggregate was estimated by dividing total fluorescence intensity by mean intensity of single cells. This could be OK if all cells were phenotypically similar throughout the aggregate, however it's quite possible that growth rate varies spatially and that cells in the center of the aggregate may be slower growing or dead, which would lower fluorescence. Do the authors have any data contradicting this? Aggregates also auto fluoresce more strongly than single cells, which could skew cell counts by signal intensity. Did the authors perform controls that calibrated or confirmed their ability to use signal intensity as a cell count measurement?

3. Results section: it would be valuable if the authors could expand their analysis to describe the range of parameter values within which their model is realistic. For example, what rates of cell division are too fast or too slow to provide the observable distributions? What cell division rates are observed in the data? Similarly, can the authors quantify fragmentation from their image data and relate that to the rates of fragmentation in their model? Could this explain why aggregates do not form in their non-aggregating wild-type strain?

4. Results and Discussion sections: A more thorough explanation about what aggregate dynamics parameters (e.g., cell division and fragmentation) makes this process like a gel is required, particularly given the title of the paper. Were the likening of bacterial aggregates a key goal of this paper, it would be stronger to have the aggregate size model explicitly compared to a soft matter model (i.e., a model of gelation transition etc.). Alternatively, the model presented in the paper has value independent of the conclusion about gels, and this value may be highlighted perhaps by reducing the emphasis on the gel-like nature of this system in the title and abstract.

5. Discussion section: The paper would be strengthened with a discussion on how different bacterial phenotypes (e.g. surface adhesiveness, heterogeneity in growth) may affect aggregate formation. It may be that mechanistic differences between some phenotypes are functionally similar for the model, enabling the model to apply across different species. For example, marine phytoplankton species have different mechanisms of cell aggregation (adhesive cell surface properties vs. mucus-mediated coagulation) [Kiørboe and Hansen, 1993]. A discussion on whether such differences can or cannot be treated by this model would help readers understand when this model is most appropriately applied.

• Thomas Kiørboe, Jørgen L.S. Hansen, Phytoplankton aggregate formation: observations of patterns and mechanisms of cell sticking and the significance of exopolymeric material, Journal of Plankton Research, Volume 15, Issue 9, 1993, Pages 993-1018, https://doi.org/10.1093/plankt/15.9.993

6. Discussion section: The suggestion that this model could be generally applied across diverse guts would be strengthened with a discussion on how the proposed model incorporates or account for these environmental factors. Fluid flow and cell/aggregate morphology, for example, are known to impact microbial aggregate formation. Could the authors discuss or speculate on how such processes affect the rates of aggregate growth, fragmentation, fusion and loss by their model?

• Kiørboe, Thomas. A Mechanistic Approach to Plankton Ecology, Princeton: Princeton University Press, 2018. https://doi.org/10.1515/9780691190310

• Jonasz Słomka, Roman Stocker, On the collision of rods in a quiescent fluid, Proceedings of the National Academy of Sciences Feb 2020, 117 (7) 3372-3374; DOI: 10.1073/pnas.1917163117

• Falkovich, G., Fouxon, A. and Stepanov, M. Acceleration of rain initiation by cloud turbulence. Nature 419, 151-154 (2002). https://doi.org/10.1038/nature00983

7. Discussion section: The authors suggest analysis of fecal aggregates from other guts (mouse, human); however, these aggregates are different from those included in their distributions as fecal aggregates are those to be expelled rather than maintained in the gut. Could the authors please describe more specifically how size distributions of expelled cells can be used to determine the rates in their model?

*Reviewer #3 (Recommendations for the authors):*

To address the goal of characterizing the distributions of gut bacterial aggregate sizes, the authors have motivated, from the ground up, an excellent first-principles-based model, and have added in complexity in layers. The model is capable of describing the aggregate size dynamics for a wide variety of gut bacteria in zebrafish. Given the specifics of the first-principles based approach used, it is plausible that it's directly applicable to gut bacteria in other animals too. Sufficient complexity is systematically added, clearly distinguishing the individual effects of each added factor on the model. Overall, the the model sufficiently explains the important features of the gut bacterial aggregate size distribution, namely, the initial power law and the final plateau.

That said, a minor issue is that the initial motivation behind building the model in this way seems somewhat unnecessary. The authors motivated the basis for the model by claiming P(size > n) ∼ n−1, using Fig. 2. But the model seems to work for any slope (depending on fragmentation rates etc). So why is the slope of -1 special?

Also, in Fig. 2, since the dashed line is separated from the actual data, it is tricky to visually compare them, and some experimental plots appear to have quite different slopes. It would be helpful if the best fit slope for the small n part is also reported.

Another minor issue: they claim that the decrease in size due to fragmentation is linked to cell division at the surface. However, after the cell divides, if only one daughter leaves the cluster then it shouldn't change the cluster's size (since size is measured in terms of numbers of cells rather than total volume). But if both daughters leave the surface, then what does it have to do with division?

These are minor issues which can be readily addressed through clear prose and presentation in the manuscript. They do not affect the model or the overall results.

---

## [Author Response]

Essential revisions:The reviewers and editors appreciate the authors' innovative study on the physical basis for spatial aggregation in gut microbial communities, a timely topic that bridges soft condensed matter physics and microbiology. The reviewers noted many strengths of the work, including the elegant combination of model system and a simple biophysical model, the mechanistic insight provided by the model, the compelling agreement between data and theoretical predictions, and the potential generality of the proposed mechanism to other systems. We also commend the authors on code that is well documented, compact, and easily accessible.We also identified several places where additional discussion would improve clarity and help the message reach a broader audience.1) We would like to see an expanded explanation of the theoretical model to make the paper more accessible to a broader audience. For example, readers with a more biological background may find a more detailed discussion of the intuitive and mechanistic basis of the model helpful (see, in particular, the suggestions from Reviewer 2, including suggestions about adding context related to aggregation in other biological systems). Similarly, readers with a more physics background might find it helpful to connect the model (and the phenomenon) more directly to previous work – for example, the authors might note that this is a variant of a preferential attachment (Yule-Simons) model, and the authors could clarify the motivation for the particular model (e.g. what is special about the slope -1?). This discussion should also include further explanation of the gel characterization, and perhaps a brief introduction to gels in the Intro section, particularly since gel is mentioned in the title and abstract.

We have added several sentences about gels to the introduction, noting both how they exemplify the link between size distributions and underlying mechanisms, and the importance of gels in living systems. We have also added sentences about preferential attachment and expulsion to the introduction, and additional explanation of the slope -1.

We provide paragraph generally introducing the modeling framework, its philosophy, and the importance of stochasticity.

We have added a paragraph to discussion about gel characterization, describing both the nature of the bacterial gel and its consequences for the scaling plots.

2) While not required, a sensitivity analysis (or similar discussion) that addresses potential sources of error in the parameter estimates would add to the work. How robust are the results to (for example) experimental errors in the measurements? In the absence of a detailed analysis, the authors could explicitly discuss any limitations imposed on the results due to these uncertainties.

We agree that a sensitivity analysis / discussion of scaling uncertainties would be useful. We have performed a sensitivity analysis; we describe this in the revised text, and also provide a new supplemental table of the best-fit values for the scaling exponents. In brief, the largest source of uncertainty in the cluster data is the mis-identification of single bacterial cells; analyzing the cumulative distribution slopes with and without the single-cell datapoints gives slopes near -1 in both cases (Table S1). Also, it is notoriously challenging to fit power laws; we consider two different methods, a linear fit to log P (size > n) vs. log n, and maximum likelihood estimation; they agree within uncertainties (see text). We also added a figure supplement addressing the potential issue of fluorescence heterogeneity within aggregates, which we find to be minor.

3) The reviewers point out several places where the text could be strengthened and clarified, and also places where a general reader might benefit from a brief mention of future extensions or future work.

We have made several changes to the text to clarify our meaning and expand our discussions. Some of these overlap the above suggestions / responses. Others are separate:

– We specify the nature of the *Vibrio* mutants;

– We clarified the relationship between the probability density and the cumulative distribution;

– We changed the Figure 3 caption;

– We added a paragraph to discussion about directly measuring aggregation and fragmentation rates as a useful future direction;

– We added a line about “chipping” fragmentation of individual bacteria, in contrast to breakup into large clusters; the latter is not observed;

– We added a line explicitly defining distribution exponent \mu in text;

– We added a line to methods specifying that each strain was tagged with GFP;

– We added a line about range of growth rates;

– We added a legend to data distributions emphasizing the “guide to the eye” slope;

– We discuss how different bacterial phenotypes can lead to different rates.

Reviewer #1 (Recommendations for the authors):I really enjoyed reading this paper, and I learned from it. My comments below are meant to help improve the work.1. Figure 1C: do the smaller clusters tend to be closer to the larger cluster? Why? Is that due to the interaction with fluid flows?

This is a good question, but we hesitate to give a quantitative answer. We have expanded the discussion to better note experimental challenges, the surmounting of which may give more direct insight into aggregation and fragmentation behaviors.

2. I think readers might find it helpful if the authors could note that this is a variant of a preferential attachment model, which I believe is better known than the Yule-Simons name.

Yes; we note this name also in the revised manuscript. We caution that the terminology is a bit confusing, since “attachment” in a network model is like growth in our model!

3. Because single gut dynamic data is available, I would have appreciated seeing some time lapse imaging of clusters merging. Are fragmentations of large clusters into smaller clusters observed?

As with #1, this is challenging to be precise about, and we hesitate to give rough data. We have expanded the discussion to better note experimental challenges, the surmounting of which may give more direct insight into aggregation and fragmentation behaviors.

4. I found the use of the term fragmentation somewhat confusing. To me what is described with single cells falling off the cluster is more akin to dispersal of propagules. Is there actual fragmentation of large clusters into medium clusters? Or is the vast majority of fragmentation due to single cells? In that case, a Leslie matrix approach might be more appropriate. If the authors could clarify whether the fragmentation of a big cluster into medium clusters occurs, that would solve the point of confusion for me. If this type of fragmentation does not occur, fragmentation might not be the best term.

This is a good point. Our terminology matches the literature from the perspective of the model, but perhaps isn’t ideal from the perspective of the organisms. (We’ve struggled with the terminology, and there probably isn’t an ideal solution.) We have added clarifying text distinguishing single cell fragmentation from “symmetric” or larger-scale break up.

5. To make the paper align better with the title, I would have found it helpful if there had been some discussion of the importance of gels in the introduction. An argument could be that it is too mundane, but if such a simple and well understood model can be mechanistically applied to a complex system such as the gut microbiome, I would argue that is a major advance. The mechanistic cases presented in the supplement do a nice job of establishing the role of individual parameters in shaping the distribution.

We agree. Please see the “essential” responses; we have added text to the introduction and discussion.

6. I struggled a bit with the fact that expulsion was not considered until later in the manuscript. If the major factors examined (including expulsion) were mentioned in the last paragraph of the introduction, it would have helped me be patient in reading.

We agree. Please see the “essential” responses; we have added a line about this to the introduction.

Reviewer #2 (Recommendations for the authors):1. Results section: a more detailed explanation in the main text of how these processes are modeled by the theory would provide added support to the generalizability of the model and enable others to clearly see how they might consider test predictions in their own systems. For example, the description of the biological interpretation and limits of the distribution exponent µ would be very valuable.

Please see above, regarding our new and more expansive introduction of the model.

2. Results section: How robust are the size distributions and model to potential error in bacterial cell enumeration? The number of cells per aggregate was estimated by dividing total fluorescence intensity by mean intensity of single cells. This could be OK if all cells were phenotypically similar throughout the aggregate, however it's quite possible that growth rate varies spatially and that cells in the center of the aggregate may be slower growing or dead, which would lower fluorescence. Do the authors have any data contradicting this? Aggregates also auto fluoresce more strongly than single cells, which could skew cell counts by signal intensity. Did the authors perform controls that calibrated or confirmed their ability to use signal intensity as a cell count measurement?

Please see our “essential” reply above on the new uncertainty / sensitivity analysis we have included.

3. Results section: it would be valuable if the authors could expand their analysis to describe the range of parameter values within which their model is realistic. For example, what rates of cell division are too fast or too slow to provide the observable distributions? What cell division rates are observed in the data? Similarly, can the authors quantify fragmentation from their image data and relate that to the rates of fragmentation in their model? Could this explain why aggregates do not form in their non-aggregating wild-type strain?

We have added text about the range of growth rates we have measured. We now discuss the range of rates for all four processes.

4. Results and Discussion sections: A more thorough explanation about what aggregate dynamics parameters (e.g., cell division and fragmentation) makes this process like a gel is required, particularly given the title of the paper. Were the likening of bacterial aggregates a key goal of this paper, it would be stronger to have the aggregate size model explicitly compared to a soft matter model (i.e., a model of gelation transition etc.). Alternatively, the model presented in the paper has value independent of the conclusion about gels, and this value may be highlighted perhaps by reducing the emphasis on the gel-like nature of this system in the title and abstract.

Done – please see the revised Discussion, and our “essential” response above.

5. Discussion section: The paper would be strengthened with a discussion on how different bacterial phenotypes (e.g. surface adhesiveness, heterogeneity in growth) may affect aggregate formation. It may be that mechanistic differences between some phenotypes are functionally similar for the model, enabling the model to apply across different species. For example, marine phytoplankton species have different mechanisms of cell aggregation (adhesive cell surface properties vs. mucus-mediated coagulation) [Kiørboe and Hansen, 1993]. A discussion on whether such differences can or cannot be treated by this model would help readers understand when this model is most appropriately applied.• Thomas Kiørboe, Jørgen L.S. Hansen, Phytoplankton aggregate formation: observations of patterns and mechanisms of cell sticking and the significance of exopolymeric material, Journal of Plankton Research, Volume 15, Issue 9, 1993, Pages 993-1018, https://doi.org/10.1093/plankt/15.9.993

These are good points. The manuscript now includes a (brief) discussion of both bacterial phenotypes (motility) and more abstract settings / morphologies such as diffusion limited aggregation and fractal forms.

6. Discussion section: The suggestion that this model could be generally applied across diverse guts would be strengthened with a discussion on how the proposed model incorporates or account for these environmental factors. Fluid flow and cell/aggregate morphology, for example, are known to impact microbial aggregate formation. Could the authors discuss or speculate on how such processes affect the rates of aggregate growth, fragmentation, fusion and loss by their model?• Kiørboe, Thomas. A Mechanistic Approach to Plankton Ecology, Princeton: Princeton University Press, 2018. https://doi.org/10.1515/9780691190310• Jonasz Słomka, Roman Stocker, On the collision of rods in a quiescent fluid, Proceedings of the National Academy of Sciences Feb 2020, 117 (7) 3372-3374; DOI: 10.1073/pnas.1917163117• Falkovich, G., Fouxon, A. and Stepanov, M. Acceleration of rain initiation by cloud turbulence. Nature 419, 151-154 (2002). https://doi.org/10.1038/nature00983

We have noted that investigating how fluid flow governs aggregation scaling would be a good topic of future study.

7. Discussion section: The authors suggest analysis of fecal aggregates from other guts (mouse, human); however, these aggregates are different from those included in their distributions as fecal aggregates are those to be expelled rather than maintained in the gut. Could the authors please describe more specifically how size distributions of expelled cells can be used to determine the rates in their model?

We don’t guarantee that this approach will work! We think our text is clear that this is a suggestion for future study and validation.

Reviewer #3 (Recommendations for the authors):To address the goal of characterizing the distributions of gut bacterial aggregate sizes, the authors have motivated, from the ground up, an excellent first-principles-based model, and have added in complexity in layers. The model is capable of describing the aggregate size dynamics for a wide variety of gut bacteria in zebrafish. Given the specifics of the first-principles based approach used, it is plausible that it's directly applicable to gut bacteria in other animals too. Sufficient complexity is systematically added, clearly distinguishing the individual effects of each added factor on the model. Overall, the the model sufficiently explains the important features of the gut bacterial aggregate size distribution, namely, the initial power law and the final plateau.That said, a minor issue is that the initial motivation behind building the model in this way seems somewhat unnecessary. The authors motivated the basis for the model by claiming P(size > n) ∼ n−1, using Fig. 2. But the model seems to work for any slope (depending on fragmentation rates etc). So why is the slope of -1 special?

We have clarified this in the Results, explaining that the slope of -1 (only) robustly emerges from growth/fragmentation

Also, in Fig. 2, since the dashed line is separated from the actual data, it is tricky to visually compare them, and some experimental plots appear to have quite different slopes. It would be helpful if the best fit slope for the small n part is also reported.

We now include all the slope values in a table.

Another minor issue: they claim that the decrease in size due to fragmentation is linked to cell division at the surface. However, after the cell divides, if only one daughter leaves the cluster then it shouldn't change the cluster's size (since size is measured in terms of numbers of cells rather than total volume). But if both daughters leave the surface, then what does it have to do with division?

We have revised the text to clarify what is meant by fragmentation.